# Third-wave cognitive behaviour therapies for weight management: systematic review and network meta-analysis protocol

Emma R Lawlor,[1] Nazrul Islam,[1] Simon J Griffin,[1,2] Andrew J Hill,[3] Carly A Hughes,[4,5] Amy L Ahern[1]

[1]MRC Epidemiology Unit, University of Cambridge, Cambridge, UK
[2]Primary Care Unit, Institute of Public Health, University of Cambridge, Cambridge, UK
[3]Division of Psychological and Social Medicine, School of Medicine, University of Leeds, Leeds, UK
[4]Fakenham Medical Practice, Norfolk, UK
[5]University of East Anglia, Norwich Medical School, Norwich, UK

**Correspondence to**
Dr Emma R Lawlor;
emma.lawlor@mrc-epid.cam.ac.uk

## ABSTRACT

**Introduction** Behavioural and cognitive behavioural programmes are commonly used to assist with weight management, but there is considerable scope to improve their effectiveness, particularly in the longer term. Third-wave cognitive behaviour therapies (CBTs) have this potential and are increasingly used. This systematic review will assess the effect of third-wave CBTs for weight management on weight, psychological and physical health outcomes in adults with overweight or obesity.

**Methods and analysis** The systematic review will be reported according to the Preferred Reporting Items for Systematic Reviews and Meta-analyses guidance. We will include studies of any third-wave CBTs focusing on weight loss or weight maintenance for adults with a body mass index (BMI) $\geq 25 \mathrm{kg/m^2}$. Eligible study designs will be randomised control trials, non-randomised trials, prospective cohort and case series. Outcomes of interest will be body weight/BMI, psychological and physical health, and adherence. We will search the following databases from inception to 16 January 2018: MEDLINE, CINAHL, Embase, Cochrane database (CENTRAL), PsycINFO, AMED, ASSIA and Web of Science. The search strategy will be based on the concepts: (1) third-wave CBTs and (2) overweight, obesity or weight management. No restrictions will be applied. We will search reference lists of relevant reviews and included articles. Two independent reviewers will screen articles for eligibility using a two-stage process. Two independent reviewers will extract data, assess risk of bias using Risk of Bias 2.0, Risk of Bias in Non-randomised studies of Interventions or Risk of Bias in Non-randomised Studies of Exposures checklist and assess quality using the Grading of Recommendations Assessment, Development and Evaluation tool. A random-effects network meta-analysis of outcomes, and sub-group analyses and meta-regression will be conducted, where data permit. If not appropriate, a narrative synthesis will be undertaken.

**Ethics and dissemination** Ethical approval is not required as no primary data will be collected. The completed systematic review will be disseminated in a peer-reviewed journal, presented at conferences and used to inform the development of a weight management programme.

**PROSPERO registration number** CRD42018088255.

### Strengths and limitations of this study

► Different third-wave cognitive behaviour therapy (CBT) modalities and delivery methods will be distinguished, enabling investigation into their comparative effectiveness.
► In addition to the direct treatment effects, indirect treatment effects will be analysed using a random-effects network meta-analysis.
► A comprehensive search strategy will be used with a large number of databases searched, no limitations applied and all prospective study designs included.
► A description of the intervention content, duration and delivery mode will be provided.
► It is anticipated that many papers will not provide sufficient details on all variables of interest, and we will be reliant on communication with corresponding authors for additional information.

## INTRODUCTION

Overweight and obesity is a major public health challenge, due to its high prevalence[1 2] and associations with reduced physical and psychological health,[3–5] as well as negative social and economic consequences.[6–8] Supporting people with overweight and obesity to achieve and maintain a healthier weight is an international priority. Most treatment approaches for overweight and obesity combine diet and physical activity advice with psychological support to make behavioural changes. Most commonly, this is a standalone treatment, although it is also an accompaniment to surgery or pharmacotherapy.[9]

There are a number of behavioural programmes that are effective for weight management in the short term, but there is substantial room for improvement in reach and effectiveness, particularly in terms of long term outcomes.[10–12] While many people are able to apply behavioural strategies in order to lose weight, it is hard to sustain these in the

face of obesogenic physical and social environments, the biological drive to maintain body weight and the habitual nature of key behaviours.

One way that the effectiveness of behavioural programmes might be improved is to identify new strategies that target psychological processes associated with better long-term weight loss and weight loss maintenance. Key among these is sustained motivation and the healthy and adaptive self-regulation of eating behaviour and emotions.[13–15] Third-wave cognitive behaviour therapies (CBTs) have been identified as a potentially useful treatment to address these important factors.

## Third-wave CBTs

Third-wave CBTs have a number of core components that distinguish these approaches from first-wave behaviour therapy and second-wave CBT, specifically an emphasis on openness, awareness and action.[16] Types of third-wave CBTs include dialectical behavioural therapy (DBT), schema therapy (ST), acceptance and commitment therapy (ACT), acceptance-based behavioural treatment (ABBT), mindfulness-based cognitive behavioural treatment (MBCT) or compassion-focused therapy (CFT).[16–18] These treatments have been associated with adaptive self-regulation and sustained motivation across a number of health domains, for example, in depression and addiction.[19–21]

There are a number of ways in which these treatments could support successful weight management. For example, fostering non-judgemental awareness of thoughts, feelings and behaviour could enable people to reduce overeating and to limit the influence of internal and external cues. Encouraging patients to experience potentially aversive internal experiences (cravings, anxiety and behavioural fatigue) and to pursue behaviour that is congruent with their goals and values could support them in adhering to a weight management plan, even when weight loss has plateaued. Fostering a compassionate attitude towards the self could help prevent discouragement following minor lapses. Mindfulness and acceptance-based therapies have also been associated with improvements in psychological outcomes related to long-term weight management, including self-regulation, dietary restraint, emotional eating, body satisfaction and mood.[22–26]

## Evidence to date

Previous systematic reviews have investigated the effect of third-wave CBTs on weight management.[22–28]

Although these reviews have consistently reported improvements for obesity-related eating behaviours and psychological outcomes, results for weight loss have been mixed. Only three reviews of interventions in people with overweight and obesity included a meta-analysis of weight or body mass index (BMI) outcomes.[22 25 26] Rogers et al's[25] meta-analysis of eight studies reported a small but significant reduction in BMI, and the 16 studies included in Carrière et al's[22] meta-analysis reported a significant

moderate effect on weight loss. However, no significant effect was found for BMI in Ruffault et al's[26] meta-analysis of nine studies. These differences may be attributable to variation in inclusion criteria and methods across the reviews.

In addition to mixed findings, there are a number of methodological limitations that constrain conclusions on the effectiveness of third-wave CBT for weight management. The majority of reviews used a small number of databases,[24–27] lacked a comprehensive search strategy[26] and conducted their database searches during or prior to 2016,[23 25–28] meaning that existing reviews will not have captured all relevant research. In addition, there has been an emphasis on mindfulness, with some reviews excluding other therapy types[22 23] and others using the term 'mindfulness' to encompass a range of third-wave CBTs. There has been a general failure in reviews to distinguish between different types of third-wave CBT or different methods of delivery, and detail on intervention content, delivery mode and intensity is lacking. This causes difficulties in assessing the comparative effectiveness of different approaches and limits their potential to inform future studies. Furthermore, little is known regarding participant adherence and attrition as only Olson and Emery[28] provided detail on this outcome. This would provide important insights into intervention fidelity and acceptability.

The present systematic review will provide a more comprehensive review of the range of third-wave CBTs that have been used for weight management. It will interrogate a large number of databases and include all prospective study designs, while acknowledging the strength of randomised controlled trials. Importantly, it will distinguish between different treatment modalities and delivery methods, by pulling both the direct head-to-head comparisons reported in earlier studies and the indirect and mixed effects within a random-effects network meta-analysis framework. Meta-regression methods will be applied to identify and/or adjust for potential source(s) of heterogeneity. While change in weight is the primary outcome, this review will also capture a range of physical and psychological health outcomes and will summarise data on intervention adherence. This review will provide important information with which to inform the development and refinement of third-wave CBTs for weight management.

## Objectives

We aim to evaluate the effectiveness of third-wave CBTs for weight management in adults with overweight or obesity.

### Primary objective

► To evaluate the effectiveness of third-wave CBTs for weight loss and weight loss maintenance in adults with overweight and obesity.

### Secondary objectives

► To evaluate the effect of third-wave CBTs for weight loss and weight loss maintenance on psychological

and physical health outcomes in adults with overweight and obesity.
► To provide a detailed description of the content, duration and delivery of interventions.
► To identify the intervention characteristics that are associated with better outcomes and adherence.

## METHODS AND ANALYSIS
### Eligibility criteria
Studies will be selected according to the criteria outlined below.

### Study designs
We will include original research articles, theses and dissertations reporting randomised control trials (RCTs), non-RCTs, prospective cohort (PC) and case series studies that report an outcome measure pre-intervention and post-intervention. No restrictions will be placed on language, year of publication or publication status.

### Participants
We will include studies of community-dwelling adult human participants (aged ≥18 years) with overweight or obesity (BMI ≥25 kg/m$^2$). Participants must be seeking assistance with weight loss or weight loss maintenance. No further restrictions will be made on gender, age, recruitment method or co-morbid conditions.

### Interventions
Studies will be included if they evaluated a third-wave CBT for the purpose of weight loss or weight loss maintenance. In terms of defining third-wave CBTs, they must be described as using DBT, ST, ACT, ABBT, MBCT or CFT. As these approaches are sometimes delivered as part of a multicomponent intervention, no restriction will be placed on the proportion of the intervention using the technique, although this may be considered in secondary analyses. Interventions will be of any duration or delivery mode. There will be no restriction placed on who delivers the intervention. If multiple arms are included in a study, any arm that meets the inclusion criteria will be included in the review. We will exclude studies that do not state that weight management is an aim of the intervention.

### Comparators
For the meta-analysis, we will include studies with no comparator, that is, single-arm pre-post studies with no control arms. We will also compare third-wave CBTs to (1) no intervention or minimal intervention, and (2) other behavioural programme(s), as reported in the studies.

### Outcomes
The primary outcomes will be:
► Body weight.
► BMI.
The secondary outcomes will be:
► Stress.
► Anxiety.

► Depression.
► Meta-cognition.
► Eating attitudes
► Eating behaviours.
► Body satisfaction.
► Quality of life.
► Blood pressure.
► Lipids.
► Glycaemia.
► Adherence to treatment.

### Timing
We will include studies that report one or more of the primary or secondary outcome measures pre-intervention and post-intervention. We will exclude studies in which the follow-up measures are less than 3 months from baseline, because of the identified need to find new approaches to improve long-term weight loss and weight loss maintenance.

### Setting
No laboratory based interventions will be included. All other settings are eligible for inclusion.

### Language
There will be no language restrictions.

### Information sources and search strategy
#### Electronic searches
We will search the following databases: CINAHL, MEDLINE, Embase, Cochrane database (CENTRAL), PsycINFO, AMED, ASSIA and Web of Science. No restrictions to the search strategies will be applied, and databases will be searched from inception to 16 January 2018. For studies published in a non-English language, appropriate employees of the university will be contacted to request a translation.

To identify studies for inclusion in this review, detailed search strategies will be developed for each electronic database by ERL, who has previous experience conducting systematic reviews, with input from a medical librarian. Other members of the systematic review team will also be consulted to ensure appropriateness of terminology and that no terms have been overlooked. The search strategy will include a number of key word and subject heading searches relating to the concepts: (1) Third-wave CBTs AND (2) Overweight, obesity or weight management. Searches in the other databases will be based on the MEDLINE search strategy (see online supplementary additional file), with modification when appropriate to take into consideration database-specific terms. Search terms for the third-wave CBTs concept will be based on a systematic review of third-wave CBTs for eating disorders by Linardon et al.[18]

#### Other search methods
To enhance literature saturation, we will manually search reference lists of all primary studies identified as eligible for inclusion in the review and previous relevant reviews.

Authors of abstracts that were identified in the database searches will be contacted to identify whether the study findings have been published elsewhere or accepted for publication. A bibliography of the included studies will be circulated to the systematic review team to ensure no other studies they are aware of are excluded.

## Study records
### Data management and selection process
We will import results from the searches into a Microsoft Excel spreadsheet, and duplicates will be removed. We will screen the study titles and abstracts to eliminate articles that clearly do not meet the inclusion criteria. There will be an initial piloting of this screening with an identical 10% of articles independently screened by two researchers to ensure consistency. If a high degree of disagreement occurs, the inclusion and exclusion criteria of the studies will be clarified through discussion with a third reviewer. Once piloting is completed, all article titles and abstracts will be independently screened by two researchers. We will obtain full-text papers where titles and abstracts are deemed to be relevant or where eligibility is unclear. The obtained full-text articles will then be independently screened by two researchers, and their eligibility will be discussed to gain consensus. Where necessary, we will seek additional information from study authors to resolve any questions about eligibility. Reasons for exclusion of articles at the full-text screening stage will be recorded. A third reviewer will resolve disagreements, if required. Reviewers will not be blinded to authors, institution or journal when screening articles. In the case of multiple articles pertaining to the same study, all articles will be included and then collated to make best use of the data. A Preferred Reporting Items for Systematic Reviews and Meta-Analyses (PRISMA)[29] flow chart will be presented showing the process of study selection.

### Data collection process
For studies that fulfil the inclusion criteria, we will extract data from articles onto a data collection form. This form will be based on the Cochrane data extraction form,[30] the Consolidated Standards of Reporting Trials 2010 statement[31] and the Template for Intervention Description and Replication checklist.[32] This will ensure that an appropriate breadth and depth of detail will be captured. The data extraction form will be piloted on three articles before it is finalised. Data will be independently extracted by two researchers, and any discrepancies will be resolved by a third reviewer. Data analysis will be conducted using statistical programme Stata V.14.2 (StataCorp. 2015, *Stata Statistical Software: Release 14*).

### Data items
Data to be extracted from the studies will include:
► General information (eg, study authors, publication year, country and source of funding).
► Study aim.
► Population description.

► Study characteristics (eg, study design, randomisation, blinding and allocation concealment).
► Participants (eg, age, sex, race/ethnicity, socioeconomic status, diagnosed condition(s), recruitment methods, sample size and weight).
► Intervention characteristics (eg, therapy type/content, mode of delivery, group or individual delivery, dose of intervention, duration of session, setting, profession who delivered the intervention and theoretical framework).
► Comparator intervention characteristics (eg, therapy type/content, mode of delivery, group or individual delivery, dose of intervention, duration of session, setting, profession who delivered the intervention and theoretical framework).
► Outcomes (eg, outcome(s) studied, whether self-reported or objectively measured, duration of follow-up, statistical analysis and intervention effect sizes).
► Adherence and attrition (eg, total number of participants at baseline and at follow-up measurements, reasons for attrition, attendance and adherence to intervention).

If studies provide data for multiple follow-up time points, data will be extracted for all time points. If multiple arms are included in a study, data from any arm that meets the inclusion criteria will be extracted.

Due to the objective of this study to provide detailed descriptions of the content of the interventions, attempts will be made to contact the corresponding author of the study to retrieve further information not provided in the article or in related publications. Study authors will also be contacted if there are any uncertainties regarding the study or missing data. If there is no response, authors will receive two email reminders. Authors will be given a 2-month timeframe to reply to this request for further information.

### Outcomes and prioritisation
The primary outcomes of interest are body weight and BMI as the aim of our review is to evaluate the effect of the third-wave CBTs on weight management. Body weight will be reported as kg. BMI will be defined as body weight (kg) divided by height (m) squared and will be reported as $kg/m^2$. We will record whether these measurements have been objectively assessed or have been obtained through participant self-report or other means.

Secondary outcomes are stress, anxiety, depression, meta-cognition, eating attitudes, eating behaviours, body satisfaction, quality of life, blood pressure, lipids, glycaemia and adherence to the third-wave CBT.

Psychosocial outcomes that have been associated with successful weight management have been included in order to understand the potential of third-wave CBTs to target these potential determinants of longer term weight control. Data for these outcomes will be extracted as reported in the study. It will be noted if a validated instrument has been used.

For blood pressure, data for systolic and diastolic blood pressure will be extracted separately, and it will be reported using the unit mm Hg. For lipids, data will be extracted for total cholesterol, low-density lipoproteins and high-density lipoproteins (HDLs) and reported using the unit mmol/L. If the ratio of total cholesterol to HDL has been presented, if possible, this will converted into the separate outcomes for total cholesterol and HDL, or the study author will be contacted for raw data. Glycaemia may be reported as glycated haemoglobin (HbA$_{1c;}$ mmol/mol or %) or glucose (random, fasting or 2 hours; mmol/L). These outcomes are important as they are closely associated with weight and are risk factors for a range of chronic diseases.

Information relevant to adherence to treatment will include number of sessions attended, length and number of times practising skills, use of resources and deviations from intervention instructions. Any information regarding participant attrition/retention, reasons for attrition and attendance at sessions will also be collected. This is important as it will enable investigation into the effect of intensity of interventions on weight management and also give insight into the acceptability of the intervention.

### Risk of bias of individual studies

Two researchers will independently assess the included studies for risk of bias. Any discrepancies will be discussed with a third reviewer to gain consensus. We will pilot this approach with three studies. Risk of bias will be assessed using the Risk of Bias 2.0 (RoB 2.0) tool,[33] the Risk of Bias in Non-randomised studies of Interventions tool[34] or Risk of Bias in Non-randomised Studies of Exposures.[35] Choice of tool will be dependent on the study design. Any other potential sources of risk of bias not covered by the tools will be noted. Reviewers will not be blinded to study authors, journal or institution. Results will be presented in a summary table.

### Meta-biases

To assess for selective outcome reporting, if a study protocol is available, the outcomes reported in the protocol and the article will be compared. If not available, the methods and results section of the article will be compared to check for any inconsistencies.

### Data synthesis

#### Intervention comparisons: direct

Standard pairwise random-effects meta-analysis will be conducted where the homogeneity of data and intervention permit, using Stata V.14.2. The head-to-head comparisons will be conducted on the outcome measures reported at specific time-points of follow-up (eg, 3 months, 6 months and 12 months). Studies that have outcomes that fall between these time-points will be combined with the closest time-point. Mean differences (for continuous data) and OR (for categorical data) and their 95% CIs will be estimated and reported.

#### Intervention comparisons: indirect and mixed

Random-effects network meta-analysis will be conducted to estimate the indirect and mixed effects using the Stata suite of commands for network meta-analysis, along with commands for visualisation and reporting of results.

#### Sensitivity analyses

Sensitivity analysis will be conducted to see the potential impact on the effect estimates of different study designs (eg, RCTs and cohort studies) by restricting the analysis to specific study types and/or by excluding one study designs at a time.

In the case of studies with two or more eligible intervention arms, dependent on the similarity of the interventions, results may be combined or be split into different groups. If different measures have been used to assess the same outcome and are sufficiently similar, they may be pooled or harmonised. If the effect measures are reported in different scales, meta-analysis will use standardised mean differences (standardised by the baseline SD value).

#### Statistical heterogeneity

The statistical heterogeneity (the portion of the variability that cannot be attributed to random error) of the studies will be tested using the I$^2$ statistic along with its 95% CI. Forest plots showing the overlap of the confidence intervals will also be provided to enable visual inspection. For the network meta-analysis, we will also report a total I$^2$ statistic and the heterogeneity variance parameter ($\tau^2$) estimated from the network meta-analysis models.

#### Analysis of subgroups or subsets

If sufficient data are available, subgroup analysis will compare:
► Type of intervention (eg, therapy type, duration, delivery and intensity).
► Comparator intervention (eg, minimal intervention or other behavioural intervention).
► Health condition of participants (eg, type 2 diabetes).
► Length of follow-up (eg, 3 months, 6 months and 12 months).
► Study design (eg, RCT, quasi-RCTs, PC and case series).
► Study quality (eg, risk of bias).
    If sufficient data on important covariates are reported in the studies, meta-regression techniques will be applied to identify and/or adjust for potential sources of heterogeneity, if applicable.

#### Narrative analysis

If meta-analysis is determined not to be appropriate due to substantial heterogeneity of studies and outcome measures that cannot be pooled, a narrative synthesis will be completed. This will be provided in the text and tables to summarise and explain the characteristics and findings of the studies.

### Missing data

It is anticipated that rates of attrition and missing data for outcome measurements may be relatively high and that studies will adopt a variety of methods for handling missing data, including using only complete cases, using all observed data, multiple imputation, baseline observation carried forward and last observation carried forward. The details of the missing data and data analysis approach will be described in the review. Our main analyses will use whichever analysis is reported in the paper. Where multiple approaches are reported, preference will be given to multiple imputation and all observed data methods. We will conduct a sensitivity analysis to evaluate whether the approach to missing data impacts on the primary outcome.

### Confidence in cumulative estimate

The quality of evidence will be assessed using the Grading of Recommendations Assessment, Development and Evaluation approach.[36] This will be used to judge studies for any methodological flaws. This assessment is suitable due to our inclusion of non-RCTs, PCs and case series design studies. Other assessments more appropriate for RCT design studies may misjudge the quality of studies using these designs.

### Patient and public involvement (PPI)

This study is part of a larger programme of research funded by a National Institute for Health Research Programme Grant for Applied Research. In the development of this research programme, we held a workshop with 22 members of Fakenham Weight Management Service, who identified the need for support with long-term weight management and who felt that psychological therapies were particularly important. The research proposal was then reviewed by three members of Fakenham Weight Management Service and the University of Cambridge PPI Panel prior to submission, then reviewed by PPI representatives on the funding panel, and revised in light of feedback. A PPI representative (Mrs Jennifer Bostock) is a member of our investigator team for this research programme; we have two PPI representatives on our programme steering committee (Mr Graham Rhodes and Mrs Norma Scullion), and we have established a Patient User Group panel for the programme. Each of these PPI representatives will contribute to the interpretation and dissemination of findings from this review and the translation of these findings into a new weight loss maintenance programme.

### ETHICS AND DISSEMINATION

The potential of third wave CBTs in weight management has been recently recognised, especially for long-term outcomes. However, evidence of their effectiveness on weight is mixed, and previous reviews have a number of limitations; our review will address these weaknesses.

Our review will distinguish between the different therapies and delivery modes, enabling investigation into their comparative effectiveness. A detailed description of the intervention content, duration and delivery mode will be provided and where possible, meta-regression will be used to identify (and/or to adjust for) the source of heterogeneity across the studies. Besides direct treatment effects using standard random-effects pairwise meta-analysis, we will also estimate the indirect treatment effects using random-effects network meta-analysis. This will allow us to include data from a range of prospective studies, not only randomised controlled trials.

Ethical approval is not required as no primary data will be collected. This systematic review will follow the PRISMA checklist.[29] It is planned that this systematic review will be published in a scientific journal, presented at relevant conferences and will be used in the development of a new weight management programme to help adults with overweight and obesity to reduce weight regain following weight loss. The findings of this review will be of interest to health professionals working with adults with overweight or obesity, researchers involved in the development, evaluation and implementation of weight management interventions and policy makers and those responsible for commissioning weight management services.

**Acknowledgements** The authors would like to thank the University of Cambridge School of Clinical Medicine librarian Eleanor Barker for assistance in developing the search strategy. We would like to thank the patient and public involvement representatives who contributed to the development of this research: Mrs Jennifer Bostock, Mr Graham Rhodes, Mrs Norma Scullion, the University of Cambridge PPI panel, and patients of Fakenham Weight Management Service.

**Contributors** ALA, AJH and SJG conceived the study, participated in the study design, provided input on methods, participated in the development of the initial search strategy and contributed to the drafting of the manuscript. ERL participated in the study design, provided input on methods, developed the initial search strategy and was responsible for drafting the manuscript. NI provided input on methods, participated in the preparation of the initial manuscript draft and reviewed drafts. CAH contributed to the drafting of the manuscript and advising and recruiting patient and public involvement. All authors critically reviewed the manuscript and approved the final version submitted for publication.

**Funding** This protocol presents independent research funded by the National Institute for Health Research (NIHR) under its Programme Grants for Applied Research Programme (Reference Number RP-PG-0216-20010). ALA and SJG are supported by the Medical Research Council (MC_UU_12015/4). SJG is an NIHR senior investigator. The University of Cambridge has received salary support in respect of SJG from the National Health Service in the East of England through the Clinical Academic Reserve.

**Disclaimer** The views expressed are those of the authors and not necessarily those of the NHS, the NIHR or the Department of Health.

**Competing interests** None declared.

**Patient consent** Not required.

**Provenance and peer review** Not commissioned; externally peer reviewed.

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
