## [Reviewer comments · BMJ Open]

ARTICLE DETAILS

TITLE (PROVISIONAL)	Third-wave cognitive behaviour therapies for weight management: Systematic review and network meta-analysis protocol.
AUTHORS	Lawlor, Emma; Islam, Nazrul; Griffin, Simon; Hill, Andrew; Hughes, Carly; Ahern, Amy

VERSION 1 – REVIEW

REVIEWER	Jiskoot Erasmus MC, The Netherlands
REVIEW RETURNED	19-Apr-2018

GENERAL COMMENTS	I am curious about the results of this review! I think it will be difficult to involve all different types of therapy for this review (very wide scope). I would narrow your scope and not include all these different types of therapy because they are so different. I would increase the inclusion criteria "12 weeks follow up from baseline" (described in timing) because you want to see the results for the long term. I don't think that 12 weeks involves long-term. Later on you describe 3, 6 and 12 months. Make it more clear.
--

REVIEWER	Margaret Allman-Farinelli The University of Sydney, Australia
REVIEW RETURNED	28-Apr-2018

GENERAL COMMENTS	Very well written protocol. My question is that using every type of study design including before and after and case studies how much confidence can we have in your findings for practice and policy. I agree that a cohort design might be appropriate for tracking long term weight gain but what will be a meaningful comparator for quantiles of weight gain and the identification of confounders. The CASP tool has considerably less rigour than the Robins-E from Cochrane. I understand that CASP will be used to assess overall body of evidence - this is quite UK specific and the international community may have more confidence in the GRADE approach.
---

VERSION 1 – AUTHOR RESPONSE

Reviewer: 1

Reviewer Name: Jiskoot

Institution and Country: Erasmus MC, The Netherlands

Comment

I am curious about the results of this review!

I think it will be difficult to involve all different types of therapy for this review (very wide scope). I would narrow your scope and not include all these different types of therapy because they are so different.

Response

We would like to thank the reviewer for their positive appraisal of the planned review. We agree that there is heterogeneity in third wave approaches and our analysis acknowledges this: while all third wave treatment approaches are included, different therapies will be analysed separately and we will describe the content of these. Indeed, we consider that including a wider range of treatment approaches than previous reviews and being specific about the content of these are key strengths of the review. Following submission of the manuscript, we have completed the article screening and identified approximately 40 articles eligible for inclusion in our review; we feel this is a manageable number and do not feel that the scope should be narrowed.

Comment

I would increase the inclusion criteria "12 weeks follow up from baseline" (described in timing) because you want to see the results for the long term. I don't think that 12 weeks involves long-term. Later on you describe 3, 6 and 12 months. Make it more clear.

Response

We agree with the reviewer that 12 weeks/3 months is not a "long term" outcome. However, our scoping of the literature prior to protocol development identified only a small number of studies with a genuinely long-term follow up. Stricter inclusion criteria would result in a much smaller pool of eligible studies, and would give us a less complete view of the evidence to date. Our meta-analysis will still allow us to isolate those studies that have genuinely long term follow-up, as we will conduct separate analyses for the outcome measures at specific follow-up time points (see Data Synthesis section on page 17) rather than having all time points pooled. This will provide important knowledge about the impact of these treatments over shorter and longer term follow-up periods. We have revised the text to ensure the same terminology regarding timing of outcome assessment and follow-up is used consistently throughout.

Reviewer: 2

Reviewer Name: Margaret Allman-Farinelli Institution and Country: The University of Sydney, Australia

Comment

Very well written protocol. My question is that using every type of study design including before and after and case studies how much confidence can we have in your findings for practice and policy. I agree that a cohort design might be appropriate for tracking long term weight gain but what will be a meaningful comparator for quantiles of weight gain and the identification of confounders.

Response

We would like to thank the reviewer for their positive appraisal of the protocol. We recognise that before and after designs and case series studies provide lower quality evidence than randomised controlled trials (and indeed not all randomised controlled trials are of equal quality), but this field is in its relative infancy and we would like to present the most complete and comprehensive review of the literature to date. In addition to direct comparisons from randomised controlled trials, the use of network meta-analysis will allow us indirectly to compare the treatment effects of single group studies to the control (and other treatment) arms of randomised controlled trials – allowing us to make a more meaningful interpretation of effect sizes seen in single group studies. Quality of all studies will be

assessed as part of the data extraction process and we will conduct sensitivity analyses that look at the impact of study type and study quality (page 17)

Comment

The CASP tool has considerably less rigour than the Robins-E from Cochrane.

Response

We thank the reviewer for this suggestion. We were unfamiliar with the ROBINS-E (which is still under development) and had chosen the CASP as it accommodated all study designs including prospective cohorts (unlike the RoB 2 of the ROBINS-I). In light of the reviewer's comment, we have decided to use the assessment with most rigour for each study design; we will use the ROBINS-I for non-randomised studies of more than one intervention, the RoB 2.0 for randomised studies and the ROBINS-E for studies where the ROBINS-I or RoB 2.0 is not appropriate. We have amended this in the text on page 16 and in the abstract and we will update our PROSPERO entry accordingly.

Comment

I understand that CASP will be used to assess overall body of evidence - this is quite UK specific and the international community may have more confidence in the GRADE approach.

Response

We agree with the reviewer that the CASP may limit the applicability to a wider, international audience. Based upon this recommendation, we have decided to use the GRADE tool to assess the quality of the evidence, instead of the CASP tool. We have amended this in the text on page 16 and in the abstract and we will update our PROSPERO entry accordingly.

VERSION 2 – REVIEW

REVIEWER	Margaret Allman-Farinelli University of Sydney, Australia
REVIEW RETURNED	31-May-2018
GENERAL COMMENTS	I am satisfied with the replies to my previous review and the manner in which they have been accommodated in the revised version. P3 8 The last sentence of the abstract has a word missing the development "OF" a weight